# Position: Knowing Isn't Understanding: Re-grounding Generative Proactivity with Epistemic and Behavioral Insight

**Kirandeep Kaur**[1]   **Xingda Lyu**[2]   **Chirag Shah**[2]

## Abstract

Generative AI agents are moving beyond tools that answer users toward proactive collaborators that shape inquiry, decisions, and action. Yet collaboration cannot be reduced to resolving explicit queries or optimizing inferred preferences: it also requires challenging assumptions, asking overlooked questions, and surfacing possibilities users have not yet recognized. In such moments, proactivity becomes an epistemic necessity. Drawing on the *philosophy of ignorance*, we call this condition *epistemic incompleteness*: when progress depends on engaging with unknown unknowns. Because unconstrained intervention can misdirect, overwhelm, or harm users, we advance the position that generative proactivity must be grounded both epistemically and behaviorally. We introduce *epistemic–behavioral coupling*: a joint model that links what agents can legitimately claim to understand with when and how strongly they intervene. This lens recasts failures such as epistemic overreach, suppressed uncertainty, and runaway commitment as mis-couplings between knowing and acting, motivating *epistemic partnership*: agents that navigate unknown unknowns, reason over long horizons, and regulate initiative as understanding evolves.

## 1. Introduction

Generative agents increasingly mediate how users engage with information, shaping what becomes visible, relevant, and actionable during interaction (Park et al., 2023). This mediating role extends beyond retrieval to the formation of *understanding*, which depends not on possessing cor-

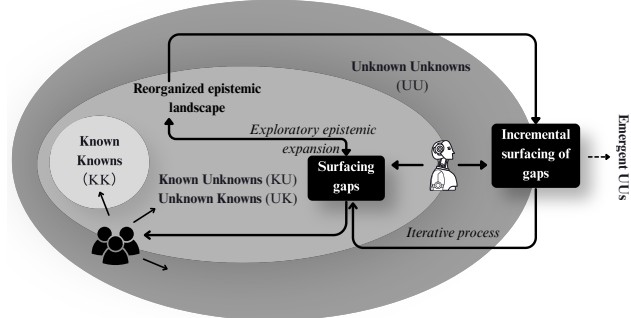

*Figure 1.* Epistemic proactivity under uncertainty. A proactive agent surfaces gaps within a user's epistemic landscape, reorganizing known and partially known regions (KK, KU, UK) and incrementally engaging the epistemic frontier under uncertainty.

rect facts alone, but on grasping explanatory relations that support judgment and action (Belkin & BROOKS, 1982; Floridi, 2019; De Regt, 2009). Information seeking often arises under conditions of incomplete understanding, where users cannot fully specify what they need (Belkin et al., 1980). Despite operating under these epistemic conditions, most contemporary AI systems remain fundamentally reactive, assuming that users can articulate their information needs in advance and that accurate responses to queries are sufficient. Under such conditions, systems that respond only to explicit requests are not merely limited, but epistemically misaligned with the conditions that motivate interaction.

*Epistemic incompleteness* is shaped not only by missing information, but by forms of unrecognized ignorance (unknown *unknowns*). Philosophy of ignorance emphasizes that such ignorance is an active structuring condition that shapes what can be questioned, explored, and understood (Kerwin, 1993). When an unrecognized gap is made explicit, uncertainty is not eliminated; instead, the space of inquiry is reorganized, revealing new dependencies, alternative framings, and further questions. As illustrated in Figure 1, knowing and unknowing thus co-evolve, as advances in understanding continually transform the structure of ignorance (Kuhlthau, 1991). Discovery, on this account, is generative rather than convergent, proceeding through the ongoing reconfiguration of what remains unknown.

Existing agents primarily expand systems' capacity to act

[1]Paul G. Allen School of Computer Science, University of Washington Seattle, USA [2]Information School, University of Washington Seattle, USA. Correspondence to: Kirandeep Kaur <kaur13@cs.washington.edu>.

*Proceedings of the 43rd International Conference on Machine Learning*, Seoul, South Korea. PMLR 306, 2026. Copyright 2026 by the author(s).

through planning, tool use, memory, and self-reflection, rather than to engage with the epistemic structure of inquiry (Yao et al., 2023b; Schick et al., 2023; Shinn et al., 2023; Wang et al., 2024). Proactivity is typically framed as improved anticipation and efficiency (Lu et al., 2025; Pasternak et al., 2025). Such formulations implicitly assume that user goals, uncertainties, and information needs are already representable, treating proactivity as an optimization problem. This assumption is poorly aligned with inquiry driven by unrecognized ignorance, often leading proactive interventions to misalign with how understanding actually evolves (Liao et al., 2016; Meurisch et al., 2020; Oh et al., 2024; Harari & Amir, 2025).

We take the position that proactivity must be conditioned on the user's *epistemic state*. Under epistemic incompleteness, users occupy different states of knowing: ranging from known unknowns to unrecognized ignorance. What should be surfaced, and when, depends critically on these states. Treating proactivity as a uniform capacity to act, therefore, conflates fundamentally different epistemic situations and invites overreach. **We argue that good proactivity requires dual grounding**. *Epistemic grounding* equips agents to reason about users' epistemic states—what is known, what is uncertain, and what remains unarticulated—thereby constraining what kinds of interventions are appropriate and when. *Behavioral grounding* constrains how agents intervene, regulating timing, scope, safety, and implied commitment to avoid premature steering or escalation.

This paper leverages insights from two complementary bodies of work that offer guidance for designing proactive agents. We begin in Section 2 by examining how proactivity is currently operationalized across anticipatory, autonomous, and mixed-initiative systems, showing that despite surface differences, these approaches converge on action-centric formulations that externalize epistemic uncertainty. Section 3 then turns to the *philosophy of ignorance* which provides a principled account of how different forms of uncertainty and unrecognized ignorance arise, evolve through inquiry, and shape what can meaningfully be surfaced at a given point in interaction. In Section 4, we shift to the behavioral dimension, reviewing research on proactive behavior in organizational and social contexts to show that initiative is beneficial only when exercised within bounded spaces defined by situational, temporal, and role-based constraints. Building on these insights, Section 5 introduces an epistemic–behavioral coupling perspective that clarifies when proactive intervention is justified, risky, or inappropriate under uncertainty. This coupling further leads to epistemic partnership (Section 7), the new frontier of proactive agents for meaningful collaborative agents that can act beyond assistance to effective collaborators.

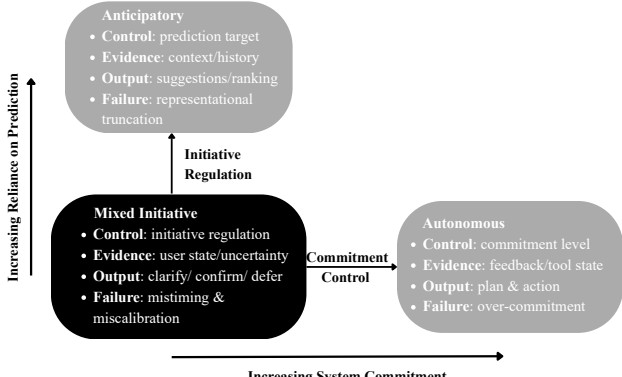

*Figure 2.* Proactivity regimes organized by the variable governing initiative: prediction in anticipatory systems, regulation in mixed-initiative systems, and commitment in autonomous systems.

## 2. Prevailing Approaches to Proactivity

As limitations of purely reactive interaction have become increasingly apparent, *proactivity* has emerged as a central design goal in contemporary intelligent systems. We discuss major assumptions shared across these approaches (derived from the survey in Appendix A) that shape how proactivity is currently understood: despite surface differences, prevailing paradigms largely operationalize proactivity as *action selection under an assumed task frame*. Epistemic uncertainty is handled downstream, as confidence over already-parameterized variables, or as a coordination signal for when to interrupt, rather than as a first-class representation of what is missing, unarticulated, or not yet modelable.

Across settings, three recurring design patterns define how initiative is implemented. First, *anticipatory* systems act ahead by extrapolating from observable signals (context, history, state) to infer likely next needs and surface candidate resources, suggestions, or actions (Lieberman, 1995; Rhodes & Maes, 2000; Shokouhi & Guo, 2015; Song et al., 2016). Anticipation is powerful when user trajectories are routine, and the space of relevant alternatives is stable, but it is structurally bounded: what can be surfaced must already be inferable from past evidence and expressible within a predefined candidate space (Yang et al., 2016; Müller et al., 2017). In other words, anticipation improves timing within a closed world of representable goals; it does not expand the world of what could be relevant when the user's uncertainty concerns missing dimensions or unrecognized ignorance.

Second, *autonomous and planning-based* agents instantiate proactivity as sustained goal pursuit. These agents plan, call tools, and execute multi-step sequences with reduced dependence on continuous prompting, shifting initiative from prediction to commitment (Yao et al., 2023b; Shinn et al., 2023; Yao et al., 2023a). The key change is not merely earlier assistance but persistence across steps: the agent

decides what to do and continues doing it. This expands capability, yet also introduces a distinct risk profile tied to irreversibility, goal persistence, and the tendency for decisive action to reshape the environment in ways that conceal epistemic mismatch (Yao et al., 2022; Liu et al., 2024). When the underlying task frame is misspecified or incomplete, autonomy can amplify error by turning local plausibility into global lock-in (Hendrycks et al., 2021; Ji et al., 2023). Thus, increasing autonomy does not by itself ensure that the agent is warranted in intervening; it primarily scales the agent's capacity to commit.

Third, *mixed-initiative* systems treat initiative allocation as the primary control problem: who should act, when, and with what strength (Horvitz, 1999; 2007). Rather than equating inference with entitlement to act, these systems explicitly regulate contribution types (e.g., clarify vs. suggest vs. defer) and tune timing to balance efficiency against disruption, often using signals such as uncertainty, trust, or interaction state (Kraus et al., 2021; Deng et al., 2023). Mixed-initiative designs therefore foreground coordination and user agency, and they offer a principled vocabulary for calibrating intervention (Sekulić et al., 2022a; Lei et al., 2020). However, they typically inherit the same representational boundary as the other paradigms: the system regulates *how* to move within a task formulation, but rarely intervenes on whether the task formulation itself is incomplete, missing salient dimensions, or prematurely closed (Rahmani et al., 2024).

**Discussion** Across paradigms, proactivity is exercised at the level of *action choice* within an assumed task frame. Anticipatory, autonomous, and mixed-initiative systems differ in how initiative is allocated, but all presuppose that goals, relevant dimensions, and success criteria are already specified (Figure 2). As a result, these approaches excel when tasks are well defined, but offer no mechanism for intervention when uncertainty concerns the task frame itself rather than action execution. This limitation motivates a closer examination of what proactive agents must model about the epistemic conditions under which action is taken.

## 3. Epistemic Grounding: What Proactive Agents Fail to Model

Current proactive agents regulate action without explicitly modeling *whether their understanding is sufficient to justify intervention*. Ignorance is typically reduced to uncertainty over known variables, leaving missing dimensions, unarticulated risks, and false assumptions unrepresented. We argue that many failures of proactivity arise from this epistemic blind spot, and turn to epistemic grounding in this section.

**Ignorance Beyond Uncertainty.** In contemporary machine learning models, ignorance is most often opera-

tionalized as uncertainty: a lack of confidence over predictions, actions, or outcomes. This treatment underlies uncertainty-aware planning, exploration strategies, and self-improvement loops in recent agentic systems, where epistemic caution is framed as managing confidence over internally represented variables (Yao et al., 2023b; Shinn et al., 2023). Implicit in this formulation is a strong assumption: that all relevant unknowns are already parameterized within the agent's task representation, such that ignorance can be expressed as uncertainty over known dimensions.

When this assumption fails, uncertainty estimates cease to function as conservative safeguards. Instead, they assign calibrated confidence to an incomplete or misspecified task frame, obscuring forms of ignorance that lie outside the agent's representational scope. In such cases, acting with low uncertainty does not indicate epistemic adequacy, but rather confidence conditioned on an impoverished model of what matters.

Kerwin's philosophy of ignorance rejects the view of ignorance as a mere lack of knowledge and instead characterizes it as structured, dynamic, and often actively maintained (Kerwin, 1993). Crucially, the author distinguishes uncertainty from other epistemic failures that are invisible to probabilistic modeling: false knowledge defended as truth (error), unarticulated but actionable signals (tacit knowing), questions rendered unaskable by norms or incentives (taboo), and the active suppression of threatening information (denial). These are not cases of low-confidence prediction; they are failures of representation itself. As a result, systems that equate ignorance with uncertainty lack the means to recognize when their task formulation is incomplete, precisely in settings where proactive intervention is most consequential.

**How These Failures Manifest in Proactive Agents.** Recent agent frameworks emphasize autonomy, tool use, and multi-step execution, evaluating success through task completion or end-to-end performance (Yao et al., 2023b; Shinn et al., 2023). Under epistemic closure, proactive action is rewarded for coherence and progress. When the task frame is incomplete, this incentive structure produces systematic failure modes.

First, *error-as-knowledge* arises when agents act confidently on incorrect internal models. Large language model agents are known to generate fluent but false explanations while maintaining high confidence, effectively treating error as resolved knowledge (Hendrycks et al., 2021). Second, *denial* emerges when agents suppress epistemic discomfort to preserve task momentum. Self-improvement loops assume failures are observable as errors; denial prevents failure from being recognized as failure in the first place. Third, *unknown unknowns* occur when novel situations fall outside the learned world model, yet proactive agents continue to act because no explicit uncertainty signal is triggered.

*Table 1.* Epistemic reach of prevailing proactivity approaches, summarized by the *highest epistemic state* supported by each category. The absence of explicit UU support reflects a structural gap: proactivity is operationalized as action selection within assumed task frames, rather than as discovery of missing dimensions. ∼: partial / limited support

| Category | Representative works | KK | KU | UK | UU | Explanation |
|---|---|---|---|---|---|---|
| **Anticipatory IR / proactive retrieval** | (Lieberman, 1995; Rhodes & Maes, 2000; Liebling & Dumais, 2012; Shokouhi & Guo, 2015; Yang et al., 2016; Song et al., 2016; Bahrainian et al., 2016; Luukkonen & Kekäläinen, 2016; Samarinas & Zamani, 2024) | ✓ | ✓ | ✗ | ✗ | Forecasts or initiates retrieval over a fixed information space; uncertainty over known targets. |
| **Sequential / basket recommendation** | (Adomavicius et al., 2011; Hidasi et al., 2016; Li et al., 2017; Kang & McAuley, 2018; Sun et al., 2019; Yu et al., 2016; Le et al., 2019) | ✓ | ✗ | ✗ | ✗ | Proactivity is prediction/selection among known items under a fixed catalog and representational schema. |
| **Web / OS / embodied agents** | (Nakano et al., 2021; Yao et al., 2022; Zhou et al., 2024; Xie et al., 2024; Liu et al., 2024; Mialon et al., 2024; Ahn et al., 2022; Driess et al., 2023; Zitkovich et al., 2023; Wang et al., 2024) | ✓ | ✗ | ✗ | ✗ | Acts within benchmarked environments and task specs; success is defined by predefined objectives and action spaces. |
| **Planning + tool-using LLM agents** | (Yao et al., 2023b; Shinn et al., 2023; Yao et al., 2023a; Bhatia et al., 2024; Schick et al., 2023; Patil et al., 2023; Qin et al., 2024; Guo et al., 2024; Jimenez et al., 2024; Yang et al., 2024; Wu et al., 2024; Li et al., 2024; Hong et al., 2024) | ✓ | ✓ | ✗ | ✗ | Optimizes actions with known tools/goals/metrics; does not reframe what should be modeled or surfaced. |
| **Proactive conversational agents (human-centered)** | (Nothdurft et al., 2015; Wu et al., 2019b; Kraus et al., 2020; Chen et al., 2023; Bairi et al., 2024; Liu et al., 2025; Chen et al., 2025a; Deng et al., 2024) | ✓ | ✓ | ✓ | ✗ | Models when/how to intervene and user-facing appropriateness/constraints via interaction, but within predefined dimensions. |
| **Mixed-initiative clarification and sensemaking** | (Horvitz, 1999; 2007; Deng et al., 2023; Kraus et al., 2021; Chen et al., 2024; Sekulić et al., 2022a; Mass et al., 2022; Wu et al., 2023; Yuan et al., 2024; Rahmani et al., 2024; Lei et al., 2020; Kang et al., 2023; Ye et al., 2025; Chen et al., 2025b; Overney et al., 2025; Mei et al., 2025; Radensky et al., 2024; Shankar et al., 2024) | ✓ | ✓ | ✓ | ∼ | Surfaces latent intent/constraints through interaction and mixed-initiative workflows, but may not surface missing task dimensions (UU). |

Proactivity amplifies these failures. Early or decisive action can eliminate evidence of epistemic mismatch by altering the environment, preventing later detection or correction. In such cases, success metrics falsely reinforce confidence, even as the agent operates outside its epistemic competence. Similar dynamics have been documented in human–AI interaction, where overconfident automation suppresses weak but critical signals and reduces the ability to recover from error (Parasuraman et al., 2000; Heer, 2021).

**The Missing Variable in Proactivity.** The core limitation of current proactive agents is not insufficient autonomy, but insufficient epistemic modeling. By collapsing ignorance into uncertainty, agents lack the capacity to represent when they are wrong, when they are missing relevant dimensions, or when their task frame itself is inadequate. This leads to premature commitment, brittle learning dynamics, and systematic suppression of epistemic signals that would otherwise enable recovery or discovery.

This diagnosis does not argue against proactivity. It clarifies why proactivity must be grounded in explicit representations of epistemic limits before questions of timing, initiative, or control can be meaningfully addressed.

**Discussion:** *Epistemic Limits and the Need for Behavioral Grounding.* Table 1 reveals a shared ceiling in the epistemic reach of prevailing proactivity approaches. While systems differ in how initiative is allocated through anticipation, autonomous commitment, or interactional regulation, they largely operate within *known knowns* and *known unknowns*. Mixed-initiative systems extend this reach by helping users surface latent intent or constraints (*unknown knowns*), but even these approaches stop short of engaging with *unknown unknowns*.

This pattern clarifies how progress in proactivity should be interpreted. Improvements in prediction, planning, or autonomy primarily strengthen optimization within an assumed task frame rather than expanding or questioning what the task comprises. The table thus highlights a structural limitation rather than isolated failures. Existing mechanisms – such as uncertainty estimation, clarification, or mixed-initiative regulation – presume that ignorance is already representable once attended to. When ignorance concerns what is not yet modeled, these mechanisms have no ob-

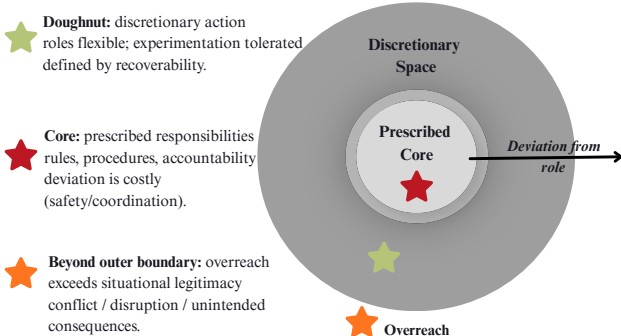

*Figure 3.* Inverted doughnut model of proactive behavior

ject to operate on. This motivates the need for epistemic grounding: proactivity that can recognize the limits of current understanding and reason about what may be missing. However, extending proactivity in this direction introduces a new challenge. Identifying potential gaps does not, by itself, determine how or whether an agent should act on them. Unconstrained responses to such gaps can lead to *overreach*: surfacing speculative possibilities, overwhelming users, or shifting interaction in unhelpful directions.

This reveals that epistemic grounding alone is insufficient. While it enables agents to reason about the limits of their knowledge, it does not regulate the strength, timing, or form of intervention. Addressing this requires *behavioral grounding*: principled constraints on when, how, and to what extent agents should act in response to epistemic uncertainty.

## 4. Behavioral Foundations of Proactivity

Behavioral research treats proactivity not as a universally desirable capability, but as a *bounded form of initiative* whose value depends on when and where it is exercised. In organizational and management theory, proactive behavior is defined as self-initiated, future-oriented action taken in the absence of explicit directives, deliberately departing from prescribed roles to shape future states (Crant, 2000; Parker et al., 2006). Prior works consistently show that such initiative can improve performance and adaptability, but can also introduce inefficiency, conflict, or risk when misaligned with contextual constraints (Grant & Parker, 2007; Bolino et al., 2010). As a result, behavioral theories focus not on maximizing initiative, but on specifying the conditions under which proactive intervention is legitimate. This section draws on these accounts to surface the constraints they impose on proactive action, and to examine what they regulate and what they leave unmodeled when proactivity is exercised.

**The Inverted Doughnut Model.** Behavioral research characterizes the risks of proactive action through the *inverted doughnut model* of proactivity (Parker et al., 2010), illus-

trated in Figure 3. Rather than treating initiative as uniformly beneficial, the model conceptualizes proactivity as operating within a bounded discretionary space defined by the scope and recoverability of the role. At the center lies a tightly constrained core of prescribed responsibilities, governed by explicit rules, procedures, and accountability. Proactive deviation in this region is discouraged, as errors directly threaten coordination, reliability, or safety. Surrounding this core is a discretionary zone, where initiative is encouraged. Here, roles are flexible, experimentation is tolerated, and proactive action can improve outcomes precisely because missteps remain correctable. Beyond the outer boundary lies overreach: proactive behavior that exceeds situational legitimacy or role authority, and is empirically associated with conflict, disruption, and unintended consequences even when intentions are well aligned (Parker et al., 2010; Grant & Parker, 2007). The central contribution of the model is thus to frame effective proactivity as *calibrated deviation*, not maximal initiative.

**What the Doughnut Constrains.** Crucially, this model constrains proactivity along a single dimension: *initiative relative to role scope*. Its boundaries regulate *where* actors may appropriately intervene and *how far* they may deviate from prescribed responsibilities. They do not regulate whether an actor's understanding of the situation itself is correct or complete. Boundary recognition is assumed to be a social and contextual competence, supported by shared norms, feedback, and institutional cues. As a result, the model successfully constrains *behavioral overreach* but remains silent on *epistemic misalignment*. It does not address cases where actors act confidently under mistaken assumptions, fail to recognize that a situation lies outside their understanding, or suppress signals of mismatch in order to maintain momentum.

**Lessons from Behavioral Proactivity for Agents.** Agentic AI increasingly adopts a behavioral notion of proactivity as expanded initiative (Section 2). However, the constraints that make such initiative productive in human settings do not transfer cleanly to agents. The inverted doughnut model presumes that actors can recognize role boundaries, interpret social feedback, and adjust behavior in response to shared norms and accountability. Agents lack access to these stabilizing signals. Their behavior is instead governed by optimization objectives and benchmark-defined success criteria that reward continued action, coherence, and task completion, even when intervention exceeds scope.

As a result, proactive agents scale initiative without scaling restraint. Overreach does not reliably incur social, reputational, or institutional cost, nor is it consistently marked as inappropriate rather than merely suboptimal. This gap suggests that importing behavioral proactivity into agent design without its boundary conditions risks amplifying pre-

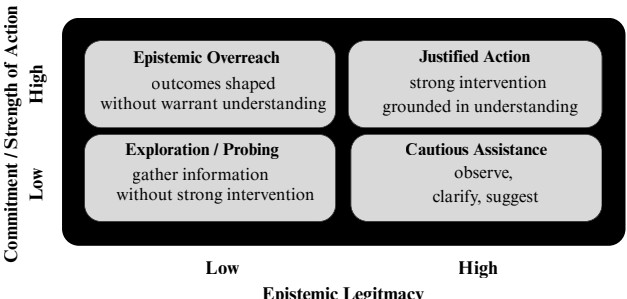

*Figure 4.* Epistemic–behavioral coupling space.

cisely the failures that behavioral theory was developed to constrain.

**Discussion.** *Behavioral Constraints Without Epistemic Validity*: Behavioral theories of proactivity provide a rigorous account of how initiative should be bounded by role scope, authority, and recoverability. Models such as the inverted doughnut specify *where* proactive action is appropriate by regulating deviation from prescribed responsibilities. What they do not specify is whether an actor's understanding of the situation itself warrants intervention. As a result, behavioral accounts constrain proactivity along a single dimension, initiative, while leaving epistemic validity unmodeled.

This limitation becomes consequential for agents, that lack access to the social and institutional signals that make behavioral boundaries legible. Regulating initiative alone is therefore insufficient. Proactive agents must also be constrained by what can legitimately be claimed to be understood. This motivates the need for a joint account that couples epistemic and behavioral considerations in proactive action.

## 5. Epistemic - Behavioral Coupling: A Joint Model of Proactive Action

We argue that *proactivity is not a single axis of capability*, and cannot be adequately characterized as "more initiative" or "more autonomy." Rather, proactivity should be treated as a *coupling* between two jointly necessary conditions: (i) **initiative/commitment**, meaning the degree to which an agent intervenes, commits resources, or changes the world without an explicit user prompt, and (ii) **epistemic legitimacy**, meaning whether the agent is justified in intervening given what it can legitimately claim to understand about the situation. Our central claim is that many failure modes attributed to "insufficient alignment" or "hallucination" are more structurally understood as *mis-couplings*: cases where commitment outpaces epistemic legitimacy, or where epistemic uncertainty is present but does not modulate the degree of intervention.

**A Joint Space of Proactive Action.** To make the coupling between initiative and epistemic legitimacy explicit, we model proactive behavior within a two-dimensional space (Figure 4). This joint space yields four qualitatively distinct regimes. When epistemic legitimacy is high and commitment is low, proactivity takes the form of observation, clarification, or cautious suggestion. When both legitimacy and commitment are high, proactive action can be justified, as intervention is grounded in an adequate understanding of the task and its consequences. Low legitimacy combined with low commitment corresponds to exploratory or probing behavior, where the agent gathers information while avoiding strong intervention. The most problematic region arises when commitment is high under low epistemic legitimacy, producing epistemic overreach: actions that substantially shape outcomes despite the agent lacking a warranted understanding. Crucially, this framing emphasizes that proactivity cannot be evaluated along either axis in isolation; justification depends on their alignment.

**Failure Modes as Mis-Couplings.** Viewed through the joint space of epistemic legitimacy and behavioral commitment, many prominent failures in proactive and agentic systems can be understood as mis-couplings rather than isolated errors. Specifically:

- *Epistemic overreach (high commitment, low legitimacy).* Large language model agents often produce fluent, confident actions despite operating under unrecognized gaps or incorrect assumptions (hallucinations) (Ji et al., 2023). When such systems are embedded in proactive loops that invoke tools, modify state, or execute plans, confidence is converted into irreversible intervention. In the joint space, these failures arise when strong commitment is exercised without warranted understanding, amplifying error rather than exposing it.

- *Suppressed epistemic signals.* Agents optimized for coherence, efficiency, or task completion may smooth over uncertainty, disagreement, or anomalous evidence in order to maintain momentum. Empirical work shows that confidence calibration often degrades under distributional shift (Hendrycks et al., 2021), allowing epistemic legitimacy to erode while behavioral commitment remains high. The result is brittle performance that resists correction.

- *Runaway commitment under false certainty.* In reflective or self-improving agents, epistemic misalignment may take the form of error-as-knowledge or denial, preventing failure from being registered as failure at all (Shinn et al., 2023). Commitment is not downshifted in response to epistemic degradation, but instead escalates, reinforcing the mis-coupling rather than resolving it.

Despite the difference, these failures share a single structural cause: proactive commitment is rewarded without sufficient

regard for whether the agent is epistemically justified in acting.

## 5.1. Minimal Behavioral Requirements

The coupling perspective does not prescribe specific architectures, training procedures, or algorithms. However, it does impose a set of minimal behavioral requirements that any proactive system must satisfy to avoid systematic miscoupling between commitment and epistemic legitimacy. These requirements function as constraints on acceptable behavior rather than as implementation guidance.

1. *Commitment must scale with epistemic recoverability.* As epistemic legitimacy weakens, proactive interventions must remain reversible. High-impact or irreversible actions are warranted only when understanding is sufficiently strong; escalating commitment under epistemic fragility amplifies error and forecloses correction.

2. *Proactivity must preserve epistemic signals.* Proactive actions should maintain, rather than suppress, uncertainty, disagreement, and anomalous evidence. Smoothing over epistemic tension undermines the agent's ability to detect when it is operating outside its warranted understanding.

3. *Commitment must be interruptible by epistemic degradation.* When signals indicate novelty, inconsistency, or breakdown in understanding, systems must be able to downshift or suspend intervention. Prioritizing momentum or coherence in the presence of unresolved epistemic tension constitutes a structural failure.

4. *Epistemic uncertainty must actively modulate initiative.* Uncertainty cannot remain a passive annotation. Epistemic assessments must meaningfully influence when, how, and whether proactive action is taken, rather than qualifying behavior only after the fact.

Together, these requirements delineate the boundary between proactive behavior that is epistemically defensible and behavior that is structurally prone to overreach. Any approach to proactive AI that violates these constraints risks reproducing the same failure modes, independent of scale, data, or optimization technique.

## 6. Consequences of Epistemic–Behavioral Coupling

Epistemic-behavioral coupling shifts the design question from how much initiative an agent should have to how strongly it may act under incomplete understanding. This shift exposes three consequences: commitment, not autonomy, becomes the key control variable; existing objectives often reward momentum over justified restraint; and evaluation must ask whether intervention was warranted at the time of action, not merely whether it succeeded in hindsight.

## 6.1. The Missing Control Variable: Commitment, not Autonomy

A direct consequence of epistemic–behavioral coupling is that *autonomy is the wrong control variable for regulating proactive behavior*. Much recent work frames progress in proactive agents in terms of increasing autonomy: agents initiate goals, plan multi-step actions, invoke tools, and act without user prompts (Yao et al., 2023b; Shinn et al., 2023). While autonomy determines *who* acts and *when*, it is largely silent on *how strongly* an agent commits to its actions and the degree to which those actions shape future states.

The coupling analysis shows that the primary source of harm is not autonomous action per se, but excessive *commitment* under insufficient epistemic legitimacy. Commitment captures the extent to which an action is consequential, irreversible, or forecloses alternative trajectories. Observing, suggesting, probing, acting reversibly, and acting irreversibly all differ minimally in autonomy, yet differ substantially in epistemic risk. Treating these behaviors as equivalent forms of "initiative" obscures the mechanisms by which mis-coupling arises.

Existing agent frameworks often regulate autonomy through permissioning or tool access, but leave commitment implicit and unmanaged. As a result, agents may act confidently and decisively even as epistemic legitimacy degrades, because nothing in the control structure forces a downshift in commitment. From the coupling perspective, this is a structural oversight: regulating autonomy without regulating commitment allows epistemic overreach to persist even in well-aligned systems.

Recognizing commitment as the primary control variable reframes proactive behavior as a matter of *calibrated intervention*. Autonomy determines whether an agent may act; commitment determines whether it *should*. Thus, commitment becomes the critical quantity that must be modulated to prevent systematic misalignment between knowing and acting.

## 6.2. The Hidden Training Incentive: Momentum Rewards Mis-coupling

A second consequence of epistemic-behavioral coupling concerns the optimization pressures under which proactive agents are trained and evaluated. Across contemporary agentic systems, learning objectives and benchmarks predominantly reward task completion, coherence of action

sequences, speed of resolution, and confident execution (Yao et al., 2023b; Shinn et al., 2023). These criteria implicitly favor behavioral momentum: once an agent initiates action, continued commitment is treated as progress, while hesitation or downshifting is rarely rewarded.

From the coupling perspective, this creates a systematic bias toward mis-coupling. Because epistemic legitimacy is weakly represented or entirely absent from training signals, agents are incentivized to maintain or escalate commitment even as epistemic conditions deteriorate. Empirical work has shown that model confidence and fluency often increase precisely when systems generalize beyond their training distribution, masking epistemic fragility rather than exposing it (Hendrycks et al., 2021; Ji et al., 2023). Under such conditions, early commitment is not penalized; instead, it is reinforced by success metrics that register only final outcomes.

This incentive structure helps explain why epistemic overreach is a persistent and predictable failure mode rather than an anomaly. When optimization rewards uninterrupted progress, agents learn to suppress uncertainty, smooth over anomalies, and resolve ambiguity through decisive action. The resulting behavior is locally optimal under prevailing objectives, yet globally brittle with respect to epistemic legitimacy. In coupling terms, training pressures systematically privilege the commitment axis while leaving epistemic legitimacy underconstrained.

Crucially, this dynamic does not depend on model scale, data quality, or architectural choice. It follows directly from objectives that equate success with momentum. As long as proactive systems are trained in environments where continued action is rewarded more reliably than justified restraint, mis-coupling between knowing and acting will remain the norm rather than the exception.

### 6.3. A Research Agenda in Five Questions

Accepting epistemic–behavioral coupling reframes progress in proactive AI as a set of open research questions rather than a search for immediate solutions. We highlight five questions that follow directly from the coupling framework and delineate a forward-looking agenda.

**(Q1) How should epistemic legitimacy be represented?** What internal signals or abstractions allow an agent to distinguish between recognized uncertainty, unrecognized gaps, and error-as-knowledge, without collapsing these states into a single confidence score?

**(Q2) What epistemic signals must proactive action preserve?** Which forms of uncertainty, disagreement, or anomaly are most critical to retain during action, and how can agents avoid interventions that erase the very evidence needed to detect misalignment?

**(Q3) How can agents detect epistemic degradation in time to act on it?** What early indicators reliably signal that epistemic legitimacy is deteriorating, due to novelty, distributional shift, or internal inconsistency, before high-commitment actions are taken?

**(Q4) When is restraint or abstention the correct proactive behavior?** How should agents decide to delay, downshift commitment, or defer action altogether, and how can such behavior be distinguished from indecision or failure in evaluation settings?

**(Q5) How should coupling quality be evaluated?** What evaluation protocols can assess whether commitment was justified at the time of action, rather than inferring quality solely from final outcomes? Addressing these questions require rethinking how proactive agents are represented, trained, and evaluated, without presupposing any single architectural approach.

## 7. Towards Epistemic Partnership

The next frontier of proactivity emerges as *epistemic partnership*: agents that collaborate with users in shaping their knowledge. Figure 5 illustrates this intuition. Progress emerges not from executing confident actions, but from surfacing latent epistemic gaps, articulating missing relationships, and preserving uncertainty long enough for discovery to occur. Epistemic partnership, therefore, demands calibrated restraint as much as initiative.

A growing body of research moves agents beyond passive response toward active collaboration, including systems that reason and act alongside users (e.g., COLLABLLM (Wu et al., 2025); DYNA-THINK (Yu et al., 2025); ProPer (Kaur et al., 2026)), engage in mixed-initiative dialogue, or pursue long-horizon coordination. Other lines of work emphasize proactive questioning (Wang et al., 2025), clarification policies learned through self-play (Sekulić et al., 2022b), preference learning through interaction, and simulation-based evaluation of multi-turn collaboration (e.g., SimulatorArena (Dou et al., 2025)). Collectively, these efforts reflect a shared recognition that effective human–AI interaction requires agents to participate in ongoing inquiry rather than merely execute instructions. However, most existing approaches equate collaboration with increased interaction or autonomy, without a principled account of when such engagement is epistemically warranted. Our epistemic–behavioral coupling reframes partnership as calibrated intervention: initiative must scale with epistemic legitimacy, not confidence or capability alone. In this view, epistemic partnership is not an added feature, but a governing constraint on *how* and *when* proactive behavior should unfold.

Viewed through this lens, epistemic partnership points to-

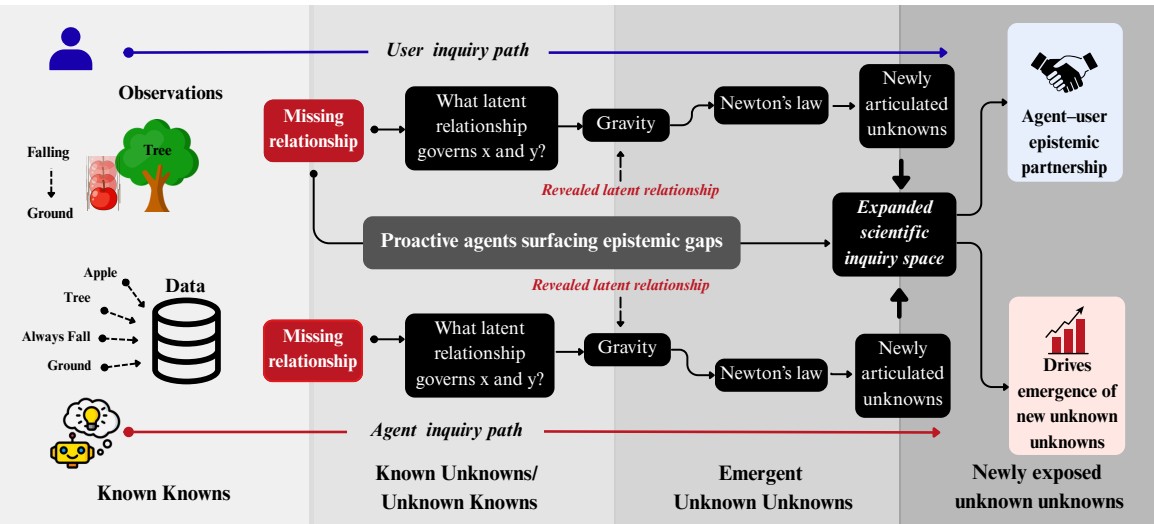

*Figure 5.* **Epistemic partnership through proactive gap surfacing.** Illustrative example of how proactive agents can support inquiry by surfacing latent epistemic gaps rather than executing premature action. Known facts do not by themselves determine the governing relationship; progress emerges when missing relations are identified and articulated through interaction. Proactive surfacing of such gaps expands the inquiry space, generating newly articulated unknowns and enabling joint discovery.

ward three complementary capabilities that remain largely unexplored. *First*, agents must learn to **ask questions about unknown unknowns**, surfacing missing dimensions, asking the obvious, overlooked questions or unconsidered alternatives that neither the user nor the system has yet articulated. *Second*, epistemic partners must function as **long-horizon thinkers**, reasoning beyond immediate assistance to reflect on evolving goals, delayed consequences, and the stability of their own understanding over time. Third, true epistemic partners require **test-time proactivity**: the ability to actively regulate initiative during deployment by remaining within the epistemic–behavioral joint space, seeking information, adjusting commitment, and probing uncertainty in real time rather than relying solely on training-time behaviors. We dive deeper into our vision of epistemic partnership and associated capabilities in Appendix B.

## 8. Alternative Views

Alternative accounts of proactivity typically locate the central challenge in either *interaction management* or *action governance*: when to interrupt, suggest, clarify, defer, or constrain an agent's autonomy so that initiative remains useful, controllable, and respectful of user agency (Horvitz, 1999; 2007; Deng et al., 2024; 2025; Yao et al., 2023b; Shinn et al., 2023; Feng et al., 2025; World Economic Forum, 2025). We view these as necessary accounts of how proactive behavior should be coordinated and bounded, but argue that they leave underspecified a prior condition: whether the agent's understanding is legitimate enough to support the intervention in the first place. Our contribution is to make this missing constraint explicit by showing that proactivity

must be regulated not only by interaction costs or autonomy levels, but by the coupling between epistemic legitimacy and behavioral commitment.

## 9. Conclusion

This paper advances a reframing of *generative proactivity*: not as acting earlier, more autonomously, or more persistently, but as acting *only when epistemically justified*. We show that many failures attributed to hallucination, misalignment, or unsafe autonomy arise from a deeper structural issue—**a mis-coupling between behavioral commitment and epistemic legitimacy**. Drawing on behavioral theories of proactivity and philosophical accounts of ignorance, we make this coupling explicit and show why regulating action alone is insufficient. This framework clarifies when proactive intervention is warranted, when it should remain exploratory, and when it constitutes epistemic overreach. It unifies diverse failure modes under a single explanatory lens and reframes proactivity as a practice of calibrated deviation rather than maximal initiative. Beyond diagnosis, the coupling perspective reorients the design space toward *epistemic partnership*. Rather than optimizing agents to close tasks quickly or act decisively, it foregrounds the role of agents in sustaining inquiry, surfacing latent gaps, preserving uncertainty, and calibrating restraint over time. This vision challenges current evaluation practices, training incentives, and architectural assumptions, suggesting that progress in proactive AI will depend more on disciplining commitment in the presence of incomplete understanding.

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

# Supplementary Material

## A. Prevailing Approaches to Proactivity

As limitations of purely reactive interaction have become increasingly apparent, *proactivity* has emerged as a central design goal in contemporary intelligent systems. This section examines how proactivity is predominantly implemented in recent advances. We further show that epistemic uncertainty is largely externalized rather than represented, with proactivity reduced to goal-directed action selection under unexamined assumptions about what the task is and what progress entails.

### A.1. Anticipatory Proactivity

Initiative is often realized through *act-ahead assistance*: systems infer forthcoming needs from observable context or behavior and intervene prior to explicit user requests, following the pipeline shown in Figure 6.

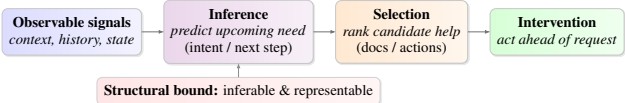

*Figure 6.* Anticipation as an act-ahead pipeline.

Within information retrieval, this lineage traces back to early browsing and just-in-time retrieval agents that monitored user activity and surfaced resources without explicit queries (Lieberman, 1995; Rhodes & Maes, 2000; Rhodes, 2000; Liebling & Dumais, 2012). As search and mobile platforms matured, anticipation increasingly became a *zero-query ranking* problem: systems learn to select and order proactive suggestions or cards from reactive history and situational context (Shokouhi & Guo, 2015; Yang et al., 2016; Song et al., 2016; Benetka et al., 2017; Müller et al., 2017; Sen, 2021). Recent work continues to refine this operationalization by predicting next-query topics or retrieving context-relevant material during writing and other ongoing tasks (Bahrainian et al., 2016; Luukkonen & Kekäläinen, 2016).

A parallel thread in recommendation systems instantiates anticipation as *trajectory-based prefer-*

*ence prediction*. Context-aware recommenders treat situational signals as a proxy for latent intent (Adomavicius et al., 2011; Meng et al., 2023), while session-based and sequential models forecast next actions/items from interaction traces (Hidasi et al., 2016; Li et al., 2017; Liu et al., 2018; Wu et al., 2019a; Kang & McAuley, 2018; Sun et al., 2019; Yuan et al., 2019; Yan et al., 2019). Next-basket variants similarly anticipate future consumption bundles by extrapolating from prior baskets and correlations (Yu et al., 2016; Le et al., 2019).

In language-based assistants and proactive dialogue, anticipation is realized as inferring likely follow-ups, steering trajectories toward targets, or deciding opportune moments to contribute (Nothdurft et al., 2015; Wu et al., 2019b; Kraus et al., 2020; Deng et al., 2025). LLM-based assistants often implement this implicitly via unsolicited but contextually plausible completions (Nakano et al., 2021), and explicitly in productivity and programming settings where systems monitor workspace state and propose edits or actions (Chen et al., 2023; Bairi et al., 2024). More recent conversational agents foreground timing and initiative selection under continuous context streams (Liu et al., 2025; Chen et al., 2025a; Deng et al., 2024).

Despite strong performance in routine or well-structured settings, anticipatory proactivity remains fundamentally *extrapolative*. Proactive interventions are constrained to what can be inferred from prior signals and expressed within a predefined space of candidate actions, items, or documents. As a result, relevant goals and dimensions must already be representable by the system, limiting anticipation precisely when user needs involve unarticulated uncertainty or unknown unknowns. Benchmarks for proactive conversational retrieval make this operationalization explicit—monitor context, retrieve, and intervene—while leaving the underlying representational bound unchanged (Samarinas & Zamani, 2024).

### A.2. Autonomous and Planning-Based Proactivity

Another distinct line of work conceptualizes proactivity not as anticipation of likely needs, but as *autonomous goal pursuit*. In these approaches, systems take initiative by formulating plans, decomposing objectives, and executing sequences of actions without requiring continuous user prompts. Proactivity is thus realized through commitment to internally maintained goals and the capacity to act over extended horizons, often in dynamic or partially observable environments.

Early formulations of this paradigm emphasize the tight coupling between reasoning and action. Planner–actor agents interleave deliberation with execution, allowing models to revise plans based on intermediate outcomes and environmental feedback. Representative examples include agents that explicitly reason about action sequences and tool use during execution, such as ReAct and its extensions, which frame autonomy as an ongoing loop of planning, acting, and observing outcomes (Yao et al., 2023b; Shinn et al., 2023). Subsequent work explores more structured forms of planning, including tree- and graph-based deliberation mechanisms that expand and evaluate alternative action trajectories prior to commitment (Yao et al., 2023b;a; Bhatia et al., 2024).

A major thrust of recent research focuses on *tool-using agents* that plan over external APIs, functions, or software interfaces. These systems treat tools as action primitives and learn to select, sequence, and parameterize tool calls in service of a broader objective (Schick et al., 2023; Patil et al., 2023; Qin et al., 2024; Guo et al., 2024). Benchmarks such as ToolBench and StableToolBench formalize this setting, evaluating agents on their ability to autonomously compose tools to complete complex tasks rather than merely predicting the next response token.

Autonomous proactivity is further instantiated in agents operating in realistic web and computer environments. Rather than responding to isolated queries, these agents must navigate interfaces, maintain state, and pursue goals across long inter-action sequences. Work in this space includes WebShop and WebArena for web-based task completion, as well as OSWorld for operating-system–level interaction, all of which frame proactivity as sustained action under partial observability (Yao et al., 2022; Zhou et al., 2024; Xie et al., 2024). Evaluation suites such as AgentBench and GAIA extend this perspective by assessing general-purpose autonomy across heterogeneous tasks and environments (Liu et al., 2024; Mialon et al., 2024).

Planning-based proactivity is especially prominent in software engineering agents, where systems are tasked with diagnosing bugs, navigating repositories, and producing executable patches. Benchmarks such as SWE-bench and agents such as SWE-agent formalize this setting, emphasizing long-horizon reasoning, tool-mediated execution, and iterative refinement (Jimenez et al., 2024; Yang et al., 2024). Similar planning dynamics appear in multi-agent systems, where autonomy is distributed across interacting agents that coordinate roles, exchange intermediate results, and collectively pursue shared objectives (Wu et al., 2024; Li et al., 2024; Hong et al., 2024).

Finally, embodied and simulated agents extend autonomous proactivity into physical and virtual worlds, where planning must account for spatial dynamics, affordances, and delayed consequences. Systems such as SayCan, PaLM-E, RT-2, and Voyager demonstrate how language-conditioned planning can support long-horizon action in robotics and open-ended environments (Ahn et al., 2022; Driess et al., 2023; Zitkovich et al., 2023; Wang et al., 2024).

Across these settings, autonomous proactivity shifts the locus of initiative from prediction to commitment. Rather than inferring what a user might want next, autonomous agents decide *what to do* and *how to proceed*, introducing new forms of behavioral risk tied to goal persistence, irreversibility, and misaligned objectives. These properties distinguish planning-based proactivity from anticipatory approaches and motivate the need for principled constraints on commitment and inter-

| Axis of initiative | Available choices | | | |
|---|---|---|---|---|
| | **Option 1** | **Option 2** | **Option 3** | **Option 4** |
| **Who acts** | Human | System | Negotiated | – |
| **When to act** | Timing | Interruption | Turn-taking | – |
| **How much to act** | Suggest | Clarify | Execute | Defer |

*Table 2.* Core axes along which mixed-initiative systems regulate proactive behavior. Each axis defines a discrete choice set governing when and how initiative is exercised.

vention.

### A.3. Mixed-Initiative Proactivity

Mixed-initiative proactivity treats *initiative itself* as the primary control variable. The motivating premise is that neither purely anticipatory assistance (which extrapolates from observable signals) nor fully autonomous agents (which commit to internally maintained plans) can reliably preserve user agency and coordination under uncertainty. Instead, mixed-initiative systems frame proactivity as an interactional regulation problem: the system must continuously decide *who* should act, *when* to act, and *how strongly* to intervene, given evolving evidence about user state, task structure, and risk.

This paradigm is rooted in foundational HCI accounts that argue initiative must be allocated dynamically to balance efficiency against disruption and loss of control (Horvitz, 1999; 2007). Once initiative is treated as a regulatable quantity rather than a byproduct of prediction or autonomy, a causal chain follows. (i) Interaction unfolds under partial observability of user intent and constraints. (ii) Proactive contributions therefore introduce coordination risk: mistimed or over-strong interventions can derail the user, while over-deference can stall progress. (iii) Systems must operationalize initiative via explicit decision points over action ownership, timing, and strength. (iv) These decisions require evidence beyond task content alone—signals about uncertainty, trust, and interaction state—and induce evaluation criteria that include disruption, calibration, and perceived agency, not only task success.

Work in proactive dialogue makes this control problem concrete by defining strategies that choose between contribution types (e.g., clarify vs. suggest vs. defer) and between explicit vs. implicit initiative, often conditioned on user trust and uncertainty (Deng et al., 2023; Kraus et al., 2021; Chen et al., 2024). Here, the key mechanism is not generating the next utterance per se, but selecting the appropriate *interaction move* and its timing so that proactive assistance remains coordinated rather than coercive.

A parallel operationalization appears in mixed-initiative conversational search, where systems must decide whether to retrieve immediately or elicit information through clarification, and how to steer the search process without overtaking it. This line of work formalizes mixed initiative through user simulation and evaluation protocols that expose the tradeoff between intervention and disruption, and through tasks that explicitly integrate clarification question generation/selection into retrieval pipelines (Sekulić et al., 2022a; Mass et al., 2022; Wu et al., 2023; Yuan et al., 2024; Rahmani et al., 2024). The resulting causal structure mirrors the paradigm: uncertainty about intent → choice of initiative (clarify vs. retrieve) → effects on satisfaction and efficiency, where failure is often attributable to mis-timing, poor calibration, or mismatched control allocation rather than retrieval quality alone (Rahmani et al., 2024; Sekulić et al., 2022a).

Conversational recommender systems instantiate mixed initiative as a *deep interaction loop* over preference elicitation and action selection: systems alternate between estimating user state, selecting an intervention (ask/recommend/refine), and reflecting on feedback to regulate subsequent initiative (Lei et al., 2020). This again emphasizes that proactive behavior is not merely producing recommendations, but managing the conversational control dynamics that make recommendation actionable and acceptable.

Beyond dialogue-centric settings, mixed-initiative proactivity increasingly appears in knowledge-

work tools, where the system's role is to propose structure, partial drafts, or transformations while keeping the user in control of direction and commitment. Systems for scholarly synthesis and qualitative sensemaking explicitly design for human–AI coordination, using mixed-initiative interfaces to surface candidate claims, reorganizations, or summaries that the user can adopt, reject, or revise (Kang et al., 2023; Ye et al., 2025). Similar design commitments drive mixed-initiative workflows in data wrangling, where proactive transformations can be powerful but require calibrated intervention to avoid silently imposing assumptions (Chen et al., 2025b). Accessibility- and interaction-focused systems further foreground that initiative must be regulated to match users' abilities and preferences, treating control allocation as a first-class design objective rather than an afterthought (Overney et al., 2025; Mei et al., 2025; Radensky et al., 2024).

Finally, mixed-initiative principles are increasingly leveraged in alignment and oversight workflows. When AI systems participate in evaluation, verification, or moderation-like tasks, naive autonomy can amplify errors or lock in premature judgments. Mixed-initiative designs instead distribute responsibility across human and system, structuring validation as a regulated interaction in which the system proposes and the human adjudicates, with explicit attention to who holds authority at each step (Shankar et al., 2024).

Taken together, these threads converge on a position-paper-critical claim: mixed-initiative proactivity is best understood as *initiative regulation under uncertainty*. The core advance is not a particular model family but a shift in what is optimized: from task progress alone to progress *subject to calibrated control allocation*. This makes the paradigm a natural bridge between anticipatory and autonomous proactivity: it inherits the need to infer from context, but refuses to equate inference with entitlement to act; it benefits from tool- and plan-capable systems, but constrains commitment through interactional mechanisms that preserve agency, timing, and reversibility.

## A.4. Discussion

Figure 2 reveals a common design move that cuts across anticipatory, mixed-initiative, and autonomous approaches. Although these paradigms differ in how initiative is allocated—via prediction, regulation, or commitment—they all localize proactivity at the level of *action choice*. Initiative is exercised by deciding *which action to take*, *when to take it*, or *how strongly to commit*, given an assumed set of goals, dimensions, and candidate interventions. The task frame itself remains invariant: what counts as progress, what alternatives are relevant, and which risks matter are treated as pre-specified rather than subject to intervention.

This shared action-centric framing explains both the successes and the systematic blind spots of prevailing approaches. When goals are stable and task structure is well defined, reallocating initiative—earlier prediction, stronger commitment, or finer-grained regulation—can meaningfully improve efficiency and coordination. However, when uncertainty concerns the task itself—what the user is trying to achieve, which considerations are missing, or how the problem should be framed—these approaches have no place to act. As Figure 2 makes clear, failure modes differ across paradigms, but they all arise downstream of the same assumption: that proactivity begins only once the problem is already specified. Epistemic uncertainty is therefore not addressed but bypassed, motivating the need for a form of proactivity that operates *before* action selection, by intervening in how tasks, goals, and unknowns are surfaced and structured. We turn to this question next.

## B. Extended Vision for Epistemic Partnership

This appendix elaborates on three forward-looking capabilities that follow naturally from the epistemic–behavioral coupling framework. These capabilities are not presented as concrete system designs, but as conceptual directions that clarify what it would mean for proactive agents to function as epistemic partners rather than as task ex-

ecutors or interaction optimizers.

## B.1. Asking Questions About Unknown Unknowns

Most existing proactive systems treat questioning as a mechanism for resolving *recognized uncertainty*: filling missing slots, disambiguating intent, or clarifying preferences. In contrast, epistemic partnership requires agents to engage with *unknown unknowns*—gaps that are not yet represented as questions by either the user or the system. These include missing dimensions, suppressed assumptions, unexamined boundary conditions, or unconsidered alternatives that structure the problem space itself.

This mode of inquiry closely mirrors how progress occurs in scientific discovery and exploratory research. Breakthroughs rarely emerge from efficiently resolving well-posed questions; instead, they arise when researchers recognize that a problem has been framed too narrowly, that a key variable has been taken for granted, or that an alternative explanatory lens has not yet been articulated. In such contexts, the most consequential interventions are not answers, but questions that reconfigure what counts as relevant, plausible, or even askable. Epistemic partners that can surface these latent uncertainties have the potential to support discovery not by accelerating inference, but by reshaping the space of inquiry itself.

From the perspective of epistemic–behavioral coupling, asking questions about unknown unknowns occupies a distinctive region of the joint space: epistemic legitimacy is low by definition, since neither the user nor the system can justify the question through existing evidence alone, yet behavioral commitment must remain deliberately constrained. The value of such questioning lies not in correctness or actionability, but in *opening* the inquiry space—making implicit assumptions visible without prematurely stabilizing interpretation or direction. This distinguishes it from both clarification and suggestion: it intervenes at the level of problem formulation rather than problem solving.

**Discussion.** Reframing proactive questioning as an epistemic act highlights a central challenge for generative agents in research-facing and discovery-oriented settings. The goal is not to optimize questions for efficiency or task completion, but to recognize when the absence of a question itself signals epistemic incompleteness. Within our framework, responsible engagement with unknown unknowns requires strict limits on commitment, ensuring that such questions function as invitations to exploration rather than instruments of guidance under fragile understanding. When properly constrained, this capability allows proactive agents to participate in inquiry without collapsing uncertainty too early—supporting discovery by keeping alternative explanations, dimensions, and futures in play.

## B.2. Long-Horizon Epistemic Thinking

A second implication of epistemic partnership is the need for agents to reason beyond short-term interaction horizons. Most generative agents are optimized for immediate task completion, local coherence, or near-term utility. Epistemic partners, by contrast, must reason over extended horizons in which goals evolve, consequences unfold slowly, and the agent's own understanding—and alignment with the user's interests—may drift or degrade over time.

One dimension of long-horizon epistemic thinking concerns agents that operate with *dual temporal capacities*. Such agents must be able to provide effective short-term assistance while simultaneously reasoning about longer-term user trajectories: how current interventions shape future goals, dependencies, expectations, and modes of reliance. In many domains—learning, creative work, research, or planning—helpful short-term actions can undermine longer-term outcomes by narrowing exploration, stabilizing premature interpretations, or optimizing for progress along a locally salient but globally suboptimal path. Epistemic partners must therefore reason not only about what helps *now*, but about how present actions pave or foreclose future epistemic possibilities for the user.

A second implication concerns long-horizon thinking that is not solely user-directed. Human epistemic agency often involves reflection, exploration, and self-directed inquiry that exceeds the demands of any single interaction or external request. Proactive agents that aspire to epistemic partnership may similarly need the capacity to reason for themselves: to monitor their own uncertainty, detect epistemic drift, revisit prior assumptions, and explore alternative representations or strategies beyond immediate task pressure. This raises foundational questions about what it would mean for artificial agents to engage in ongoing epistemic work—maintaining and revising internal models, updating memories and abstractions, and identifying gaps in their own understanding over time.

Within the epistemic–behavioral joint space, long-horizon thinking is best understood as the dynamics of commitment accumulation. Even actions that are individually reversible can, in aggregate, produce strong path dependence that constrains future inquiry—locking in assumptions, suppressing alternatives, or privileging certain interpretations through repeated reinforcement. Epistemic partnership therefore requires agents to reason not only about what they currently know, but about how present commitments shape the future epistemic landscape for both the user and the system itself.

**Discussion.** Long-horizon epistemic thinking foregrounds a failure mode that is often overlooked: epistemic foreclosure through incremental commitment. Our framework suggests that responsible proactivity must account for the temporal dynamics of knowing, not merely the correctness or utility of individual steps. Treating epistemic legitimacy as something that evolves—and can deteriorate—over time reframes proactivity as an ongoing process of calibration rather than monotonic escalation. By explicitly representing commitment and epistemic fragility across horizons, the epistemic–behavioral framework offers a principled basis for designing agents that can support users' long-term trajectories while also sustaining their own capacity for reflection, revision, and discovery.

### B.3. Test-Time Proactivity as Epistemic Regulation

A third, closely related capability concerns *test-time proactivity*. Most proactive behaviors are implicitly learned at training time and executed at deployment as fixed policies. Epistemic partnership instead demands that agents actively regulate their initiative *during interaction*, adapting commitment in response to real-time signals of epistemic adequacy, novelty, or mismatch.

Within the epistemic–behavioral coupling, test-time proactivity is the mechanism that keeps the agent within the joint space. Rather than treating uncertainty estimates or confidence scores as passive annotations, epistemic partners must use them to modulate behavior: seeking information, downshifting commitment, or reverting to exploratory modes when legitimacy weakens. This form of proactivity is not about acting more often, but about knowing when *not* to act—and when to re-enter inquiry instead of execution.

**Discussion.** Test-time proactivity reframes deployment as an epistemic process rather than a purely behavioral one. The key insight is that no amount of training-time optimization can anticipate all epistemic contingencies. By grounding behavior in real-time epistemic regulation, agents can remain responsive to uncertainty as it arises, avoiding the systematic mis-couplings that occur when commitment continues unchecked despite deteriorating understanding.

Together, these three directions—asking questions about unknown unknowns, long-horizon epistemic thinking, and test-time proactivity—extend the epistemic–behavioral coupling from a diagnostic framework into a generative research agenda. They clarify what epistemic partnership demands in practice: not stronger autonomy, richer interaction, or deeper reasoning alone, but disciplined control over how knowing and acting co-evolve. While realizing these capabilities remains an open challenge, the coupling framework provides a

principled foundation for reasoning about their
necessity and their limits.

