# OpenReview forum: "Position: Knowing Isn’t Understanding: Re-grounding Generative Proactivity with Epistemic and Behavioral Insight"
_ICML.cc/2026/Position_Paper_Track — ICML 2026 Position Paper Track regular_

### Official Review · Reviewer_Q521 · 2026-03-11

**Significance:** 3
**Argument Clarity:** 3
**Rating:** 5
**Confidence:** 4

**Questions:**

Please see weaknesses above.

**Alternative Views Section:**

Yes

**Compliance With Llm Reviewing Policy A Conservative:**

Affirmed.

**Discussion Potential:**

3

**Final Justification:**

As I reflect on the author's responses to my concerns and review the other reviews and comments, I find my overall support for this paper to be wavering. I do not think that the authors have provided sufficient evidence for their characterization of the core phenomenon. This makes me concerned that the paper is setting off on the wrong foot and therefore not staking out a position that will lead to fruitful discussion. I will stick with my current overall rating of 5. but I do not anymore have complete confidence that this is the right choice.

**Paper Summary:**

This position paper offers a framework for thinking about how AI systems should be *proactive*, that is, seek to anticipate user needs. The paper argues that proactive behavior should be grounded in the user's needs and knowledge state, and it should be regulated by the AI's own capabilities and the broader context of safety, information secrurity, etc. The paper includes a quite comprehensive overview of the literature on proactive behaviors in AI, and it describes a flexible framework for thinking about proactivity as joint action between the user and the AI. The overall message seems to be that current proactive behaviors from AIs are kind of haphazard and therefore often fail in their basic goals.

**Position:**

Yes

**Position In Title:**

Yes

**Related Work:**

4

**Strengths And Weaknesses:**

Strengths

I am impressed by this paper. It contains an extensive amount of scholarship drawing on multiple literatures, and it offers concepts and ways of thinking that seem like they will be very helpful when it comes to thinking about proactivity in AI agents. I am overall in favor of publication and I expect that I will cite this work in the context Next up my research on communication and interaction, as well as my research in AI.

Weaknesses

1. In my experience, the top LLM-based AI services at present are extremely proactive, and in ways that I generally find helpful. When I ask technical questions, they often provide a range of different options for the answer, and they often flag presumptions of the named methods that might be relevant (distributional facts about the data, etc.). Very often this is much more information than I was expecting, which might be a failure of epistemic grounding, but it seems relatively harmless to be doing this in the way of a very helpful interlocutor. Thus, I think I disagree with the core claim in the intro that "most contemporary AI systems remain fundamentally reactive". All of this behavior seems proactive in the context of the interaction I initiated.

2. I find the core concepts offered in the paper (the different kinds of proactivity, the joint model of interaction, the categories of epistemic reach) to be useful, but I am concerned that they won't be actionable for AI system designers in the current moment. Even for the current paper, I think the logical conclusion of all these observations is that epistemic grounding and behavioral grounding are incredibly flexible notions that will vary not only between users, but also between user interactions and perhaps even between between turns in the conversation. It's thus not clear to me what it means to take action on the ideas in the paper, behind using them as high-level guidance for designing post-training regimes, working with user context, etc. In the end, nature will not divide up into these categories, nor will current AI system design practices allow us to treat all of these notions as fixed guidelines.

3. It's really surprising to me that there's not more discussion of user preferences in the paper. Perhaps this is meant to be implicit in the notion of epistemic proactivity, but to me this seems like a separate idea. I did wonder whether my desire for more talk of preferences reflects the fact that I read the paper as primarily about being interactions between humans and AIs, whereas much of the text of the paper seems to be trying to operate at a higher level in which AI systems might be doing very diverse tasks, perhaps not even interactional ones.

**Support:**

4

---

> ### Author Rebuttal · Authors · 2026-03-29
>
> What stood out to us in the review is the emphasis on how our framework can inform thinking about *communication and interaction in AI systems*. We thank the reviewer for positive feedback and are glad that the synthesis across literatures and the proposed concepts are seen as useful for shaping future work in this space.
>
> ### **Weaknesses**
> >  *Perceived proactivity in current LLM behavior.*
>
> We thank the reviewer for eliciting a very important behavior of current LLMs. LLMs often provide rich, multi-faceted responses that feel proactive and are frequently helpful, and we do not dispute their value.
>
> Our claim is more specific: these systems remain largely reactive at the level of *problem framing*, even when they appear proactive in *response generation*. For example, **when a user asks a technical question**, the system may offer multiple solutions or highlight assumptions within that frame. While useful, this assumes the problem is already well specified. In practice, users often pose queries precisely because of underlying knowledge gaps[1.]. They may be missing key context or may not yet know what to ask. *Existing models rely on the user to ask the right questions, which disadvantages those with low information literacy.*
>
> This reliance on the user to frame the problem can limit effectiveness, especially in settings like **exploratory search, brainstorming, or complex decision-making**, where the goal is to refine the problem itself. In such cases, more effective interaction requires the system to actively explore beyond mere engagement while considering user's epistemic incompleteness and the behavioral requirements.
>
> In our formulation, proactivity involves reasoning about what is not yet represented, not just expanding what is given. As discussed in **Section 2**, existing paradigms (anticipatory, autonomous, mixed-initiative) largely optimize behavior within a fixed frame, rather than reasoning about whether that frame is itself incomplete. We will revise the paper to make this distinction clearer.
>
> [1.] Nicholas J. Belkin, Robert N. Oddy, and Helen M. Brooks. Anomalous states of knowledge as a basis for information retrieval. The Canadian Journal of Information Science, 5:133–143, 1980.
>
> > *Operationalizing proactivity under dynamic user and interaction contexts.*
>
> We agree that epistemic and behavioral states are dynamic and vary across users and interactions. Our goal is not to treat them as fixed categories, but to elicit actionable **control variables** for system behavior.
>
> A central issue in current systems is that uncertainty about known factors is often treated the same as situations where important aspects of the problem are simply not represented. Our framework focuses on this latter case, *when the system may be missing key variables, assumptions, or user constraints*, which calls for different signals and responses.
>
> In practice, while developing a true understanding of epistemic states of user can be challenging, our framework leads to simple design rules for operationalizing: when the system detects incomplete understanding (e.g., reliance on unverified assumptions or missing context), it should reduce commitment by asking clarifying questions or offering tentative suggestions; when its understanding is stable, it can act more decisively. These behaviors can be implemented using signals such as inconsistency across outputs, need for clarification, or sensitivity to assumptions.
>
> This perspective is reflected in **Section 5** (commitment scaling, reversibility, interruptibility) and further connected to training and evaluation in **Appendix B**, for example by rewarding actions that are justified given available information and evaluating whether interventions were appropriate at the time.
>
> > *Discussion on user preferences.*
>
> User preferences are indeed an important part of proactive systems, and our intent is to incorporate them through behavioral grounding rather than treat them as a separate axis.
>
> In our framework, **epistemic grounding** focuses on what the user knows or may be missing (e.g., their awareness, memory, or gaps in understanding), while **behavioral grounding** governs what is appropriate given the user and context, such as how much proactivity is desirable, what level of intervention is acceptable, and what may be intrusive or irrelevant. In this sense, behavioral grounding naturally captures user preferences as they relate to action.
>
> We also note the reviewer’s observation regarding interaction-centric settings. While the paper operates at a more general level, these ideas directly apply to human–AI interaction, where preferences can vary across users, tasks, and even over the course of a conversation. We will revise the paper to make this connection more explicit and clarify how preference-sensitive behavior fits well within the framework.

---

> > ### Author Rebuttal · Reviewer_Q521 · 2026-04-01
> >
> > I am okay with sticking with my currently positive score, but I do still have the concerns that I raised in my original review. For example, the author response says, "Existing models rely on the user to ask the right questions, which disadvantages those with low information literacy." This is simply not my own experience. I'm very open-minded about the fact that my experiences are not the norm, but I think I need the authors to offer me the supporting evidence for this claim as opposed to simply asserting it. It feels like a characterization from four years ago.
> >
> > I also think it is a mistake for the authors to assume that they can reduce preferences to the concepts in the paper. There is no reason to believe that as far as I know. The paper itself is not committed to this position though, so I'll set this aside, but it is something that the authors seem to be saying in their response.

---

### Official Review · Reviewer_bvPp · 2026-03-12

**Significance:** 3
**Argument Clarity:** 3
**Rating:** 4
**Confidence:** 2

**Questions:**

1. The paper introduces generative proactivity as a core concept. Could the authors further elaborate on how users' epistemic gaps can be identified in practical systems? Specifically, are there actionable metrics or signals to assess the user's current knowledge state and potential information deficits?

2. When proposing a proactive agent, the system must determine when and how to intervene in the user's workflow. Have the authors considered specific strategies or mechanisms for triggering these interventions, such as those based on user behavioral patterns, task stages, or expected information gain?

**Alternative Views Section:**

Yes

**Compliance With Llm Reviewing Policy A Conservative:**

Affirmed.

**Discussion Potential:**

3

**Final Justification:**

The rebuttal solves my question and hence I decide to increase my score.

**Paper Summary:**

The paper examines how generative AI systems should understand and respond to situations in which users’ knowledge and intentions are incomplete. It argues that many current AI systems treat understanding as the ability to answer explicit user queries, assuming that users can clearly specify what information they need. The authors contend that this assumption overlooks a common condition of epistemic incompleteness, where users may not know what they do not know and therefore cannot formulate the relevant questions. Under this condition, systems that only react to explicit prompts may fail to support effective decision-making or learning. To address this limitation, the paper introduces the concept of generative proactivity, which describes the capacity of AI systems to identify potential gaps in a user’s knowledge and to suggest relevant information, questions, or perspectives that the user may not have explicitly requested. The authors propose that such proactive behavior can help users explore unknown aspects of a problem and extend their understanding beyond the information explicitly available in a prompt. The paper further discusses how proactive agent could be incorporated into human–AI interaction by considering users’ epistemic states and by structuring system behavior so that suggestions are relevant and appropriately timed.

**Position:**

Yes

**Position In Title:**

Yes

**Related Work:**

2

**Strengths And Weaknesses:**

**Strengths**:
This paper presents a compelling conceptual framework by addressing "epistemic incompleteness", scenarios where users cannot fully articulate their knowledge gaps. By introducing the concept of "generative proactivity," the authors effectively challenge the prevailing reactive, prompt-based paradigm of AI systems, advocating instead for models that actively help users explore unknown aspects of a problem. Ultimately, the work successfully frames a novel interaction paradigm and motivates a valuable research agenda aimed at enhancing human-AI collaboration through controlled, proactive agent.

**Weaknesses**:
Despite its strong conceptual foundation, the paper's arguments rely too heavily on high-level reasoning, lacking the concrete validation or formal mechanisms needed to reliably infer users' epistemic states and time interventions. Furthermore, the proposed research agenda is underspecified, missing clear evaluation metrics, design principles, or measurable criteria for identifying epistemic gaps. Finally, this paper requires a more rigorous comparison with existing literature on mixed-initiative interfaces and proactive decision-support systems to properly contextualize the work and highlight its specific technical novelty.

**Support:**

2

---

> ### Author Rebuttal · Authors · 2026-03-28
>
> We thank the reviewer for insightful questions and are encouraged that the paper’s core ideas (the notion of **epistemic incompleteness** and **generative proactivitity**), are recognized as a *compelling* and *valuable* direction for improving human-AI interaction.
>
> ### **Questions**
>
> > *Identifying epistemic gaps in practical systems*
>
> In practice, epistemic gaps are not directly observable but can be inferred from proxy signals that also serve as measurable criteria. These include: (i) **uncertainty under perturbation**, where inconsistent outputs across equivalent inputs indicate unstable understanding; (ii) **clarification need**, where interaction patterns reveal missing information or evolving user constraints; and (iii) **assumption sensitivity**, where an intervention depends on unverified or implicit assumptions.
> We will revise the paper to make these signals and their role in detecting epistemic gaps more explicit.
>
> > *Strategies for triggering interventions*
>
> Determining when and how to intervene follows directly from the design principles in **Section 5**. In our work, intervention policies are governed by epistemic confidence and commitment scaling (Section 5.1), rather than fixed heuristics. Concretely, **signals** such as epistemic uncertainty or instability indicate low confidence and lead to reduced or deferred interventions, while stable and well-supported inferences justify stronger commitments. Similarly, **task progression** informs the form of intervention, with earlier stages favoring exploratory or clarifying actions and later stages permitting higher-commitment decisions, consistent with the commitment-scaling principle. **Interaction patterns** (e.g., hesitation or repeated corrections) further act as indicators of epistemic degradation (interruptibility principle discussed in Section 5). As outlined in Appendix B.3, these decisions can be framed in terms of expected information gain, where intervention is triggered when the anticipated reduction in epistemic uncertainty justifies the cost of action.
>
> ### **Weaknesses**
> > *Reliability in inferring users' epistemic states and time interventions.*
>
> While the paper may appear high-level, we intend to position the work at the level of problem formulation, identifying epistemic legitimacy as a missing variable in proactivity.  To ground this, reliability in our position depends on correctly inferring the user’s knowledge state and calibrating interventions accordingly.
>
> For example, consider a query *“Why does my model overfit?”*. If the system suggests advanced techniques (e.g., dropout tuning), it may be assuming knowledge the user does not have. Instead, **interaction cues**, **persona or memory retrieval**, and **follow-up questions** can help distinguish whether the user lacks foundational understanding or seeks deeper optimization. When this inferred epistemic state is uncertain, the system should reduce commitment and seek clarification (Q1). This same notion of epistemic stability governs the timing of interventions (Q2). We further connect this to training and evaluation in **Appendix B.2–B.3**.
>
> > *Clarification on research agenda, evaluation metrics, and design principles.*
>
> We agree that the research agenda can be made more explicit. Our intent, however, is to move beyond prescribing a fixed metric or framework and instead elicit a novel interdisciplinary perspective to proactivity. **Section 5** of our paper introduces key design principles, including commitment scaling with epistemic confidence, preference for reversible actions under uncertainty, and interruptibility under epistemic degradation. Similarly, **Appendix B** outlines evaluation directions. Particularly, **Appendix B.3** formulates this as a set of research questions: (RQ1) how epistemic legitimacy can be represented or inferred, (RQ2) how epistemic degradation can be detected and acted upon, and (RQ3) how proactive behavior should be evaluated beyond outcome-based success.
>
> > *Comparison with mixed-initiative interfaces and proactive decision-support systems.*
>
> Prior work in mixed-initiative interfaces [1] and proactive decision-support systems [2] primarily focuses on how control is shared (e.g., when the system vs. the user should act) or how outcomes can be improved through intervention. In contrast, our work introduces **epistemic legitimacy as a first-class variable**. We note that **Section 2** begins to draw this comparison by analyzing anticipatory, autonomous, and mixed-initiative paradigms and highlighting a shared failure mode -- the absence of explicit epistemic modeling. We agree, however, that this distinction can be made sharper. We will revise Section 2 and the introduction to more directly contrast our framework.
>
> [1] E. Horvitz. Principles of Mixed-Initiative User Interfaces. Proc. CHI, 1999.
> [2] D. S. Weld and G. Bansal. The Challenge of Crafting Intelligible Intelligence. Communications of the ACM, 2019.

---

> > ### Author Rebuttal · Reviewer_bvPp · 2026-04-04
> >
> > I think the rebuttal solves my question

---

### Official Review · Reviewer_yLH4 · 2026-03-12

**Significance:** 3
**Argument Clarity:** 3
**Rating:** 4
**Confidence:** 3

**Questions:**

See Weaknesses (2).

**Alternative Views Section:**

Yes

**Compliance With Llm Reviewing Policy A Conservative:**

Affirmed.

**Discussion Potential:**

3

**Final Justification:**

The additional clarification, especially regarding Weakness (1), which has improved my understanding. I have increased my score by one point.

**Paper Summary:**

The paper discusses limitations of current proactive AI agents, arguing that many systems treat ignorance primarily as uncertainty over already represented variables. It introduces an epistemic–behavioral perspective on proactivity, suggesting that proactive intervention should depend not only on the degree of initiative but also on whether the agent is epistemically justified in acting. The paper draws on the philosophy of ignorance and behavioral research on proactivity to develop a conceptual framework for analyzing proactive behavior in agent systems.

**Position:**

Yes

**Position In Title:**

Yes

**Related Work:**

2

**Strengths And Weaknesses:**

Strengths
- (1) Clear identification of epistemic limitations in existing proactive agents. The paper clearly points out epistemic limitations in existing proactive agents. By distinguishing between different forms of ignorance—such as unknown unknowns, tacit knowledge, denial, and representational failures—it clearly reveals important blind spots in current proactive system design.

Weaknesses
- (1) While the paper presents a clear conceptual argument about epistemic–behavioral coupling (Section 5), the position is largely supported by philosophical reasoning rather than empirical evidence or concrete system analysis. In particular, the joint model of proactive action and the “minimal behavioral requirements” are introduced as normative principles, but the paper does not show how key concepts such as epistemic legitimacy could be operationalized or measured in current ML systems. As a result, the argument remains largely conceptual, and its practical implications for system design are somewhat unclear.
- (2) The paper offers a useful conceptual synthesis, but the novelty of the proposed perspective relative to existing discussions on robust AI, AI safety, and decision theory on uncertainty (e.g., behavioral economics) remains somewhat unclear.

**Support:**

2

---

> ### Author Rebuttal · Authors · 2026-03-29
>
> We appreciate the reviewer’s engagement with the paper and their recognition of the **role of epistemic limitations**, particularly the *distinctions between different forms of ignorance*, in shaping proactive behavior.
>
> ### **Weakness 1: Practical Implications**
> >  *Philosophical nature of Epistemic- Behavioral Coupling.*
>
> We acknowledge the reviewer’s observation that coupling is presented at a conceptual level.  At the same time, **Section 5** connects this idea to design through principles such as adjusting action strength based on confidence, preferring reversible actions under uncertainty, and interrupting when confidence drops. These can be interpreted as constraints on agent behavior. Our goal is to elicit how a system’s and user's knowledge (and gaps in it) should determine whether and how an agent acts, which is not clearly modeled in proactive agents.
>
> We further note that **Appendix B** (B.2,B.3) outlines how these ideas relate to training and evaluation. Training objectives can discourage overconfident actions under incomplete information and reward clarification or deferral when uncertainty is high. Similarly, evaluation can assess whether actions were justified given the information available, rather than based only on final outcomes. We will revise and make these connections clearer.
>
> >  *Operationalizing epistemic legitimacy.*
>
> We agree that the operationalization of epistemic legitimacy can be made more explicit. In practice, it can be approximated through observable signals of incomplete or uncertain knowledge. For example, inconsistency across similar inputs, dependence on unverified assumptions, or repeated need for clarification can indicate gaps in understanding.
> These signals can be used to guide behavior, for instance, reducing action strength, asking clarifying questions, or delaying decisions when confidence is low. These signals can also be quantified and used within ML systems, e,g. , by measuring output consistency, tracking clarification frequency, or estimating dependence on missing inputs. Such signals can guide both training ( discouraging overconfident actions under uncertainty through RL) and evaluation (assessing whether actions were justified given available information). We will revise the paper to make these measurement strategies more explicit.
>
> >  *Practical implications for system design.*
>
> While presented conceptually, the framework maps to concrete ML mechanisms. A key challenge in current systems is that **uncertainty (over known variables)** is often conflated with **ignorance (missing or unmodeled variables)**. Our framework focuses on the latter, which requires different signals and behaviors.
>
> In **Section 5**, behavioral principles (e.g., deferring, clarifying, or committing) can be implemented as policy decisions conditioned on epistemic signals. In **Appendix B (B.2–B.3)**, we outline how these signals can be integrated into learning. For instance, model uncertainty (e.g., output variance) captures known uncertainty, while signals such as sensitivity to missing inputs, inconsistent assumptions, or repeated clarification indicate epistemic gaps. These can be used as features in learned policies (e.g., RL or bandit-style) or as reward signals to discourage overconfident actions and encourage clarification. Evaluation can similarly assess whether actions were justified given the information available at the time. We will revise the paper to make these distinctions and ML connections more explicit.
>
> ### **Weakness 2: Novelty relative to AI safety and decision theory**
> We truly appreciate this important observation. The distinction we aim to highlight is not only in how uncertainty is handled, but also in the **setting considered**: while much prior work focuses on uncertainty within a defined problem space, our work emphasizes cases of epistemic incompleteness, where relevant variables or assumptions may be missing.
>
> In practice, many failures in current systems are not only about harmful outputs, but also about overconfident or poorly timed behavior, (e.g. providing detailed but misaligned suggestions when the user’s query is underspecified, or confidently answering when the system lacks sufficient context). These issues can lead to user frustration and reduced trust in human-AI interaction.
>
> Our contribution is to address this gap through an epistemic–behavioral coupling, where behavior is governed by the system’s knowledge state. This shifts the focus from only how to act safely under uncertainty to also whether and how strongly to act given incomplete understanding, which we believe is essential for human–AI collaboration. Our work also supports behavior under epistemic incompleteness, e.g. , proactively identifying gaps in a user’s understanding and generating clarifying questions or alternative perspectives, as well as enabling more exploratory behavior to expand the system’s own knowledge space **(Section 6, Towards Epistemic Partnership)**

---

> > ### Author Rebuttal · Reviewer_yLH4 · 2026-04-03
> >
> > Thank you for the additional clarification, especially regarding Weakness (1), which has improved my understanding. I have increased my score by one point.

---

### Official Review · Reviewer_DPN2 · 2026-03-14

**Significance:** 3
**Argument Clarity:** 3
**Rating:** 4
**Confidence:** 4

**Questions:**

See Weakness above.

**Alternative Views Section:**

Yes

**Compliance With Llm Reviewing Policy A Conservative:**

Affirmed.

**Discussion Potential:**

4

**Paper Summary:**

This paper argues that generative AI agents often equate understanding with resolving explicit queries. This assumption leads to failures when users lack awareness of missing or risky information. The text introduces "epistemic incompleteness" to describe conditions where progress depends on engaging with unknown unknowns. Current proactivity approaches are critiqued for treating initiative as action optimization within assumed task frames. Instead, the authors propose dual grounding: epistemic grounding for reasoning about users' knowledge states, and behavioral grounding for constraining intervention timing and scope. Drawing on philosophy of ignorance and behavioral proactivity research, the work presents an epistemic-behavioral coupling framework. It advocates for "epistemic partnership" as a new direction for proactive agents.

**Position:**

Yes

**Position In Title:**

Yes

**Related Work:**

3

**Strengths And Weaknesses:**

**Strengths**

- The paper presents a fresh conceptual reframing by combining Kerwin's philosophy of ignorance with the inverted doughnut model from behavioral science. This cross-disciplinary foundation is uncommon in AI agent research and offers useful new terms for diagnosing proactive failures.

- The systematic critique in the paper clearly identifies a shared limitation across anticipatory, autonomous, and mixed-initiative paradigms. All treat proactivity as action selection within assumed task frames. This helps the community see a structural blind spot rather than isolated technical issues.

- The four minimal behavioral requirements and the joint epistemic-behavioral space provide concrete and architecture-agnostic constraints. Examples include scaling commitment with epistemic recoverability and preserving uncertainty signals. These could directly guide the design of safer and more reflective proactive systems.
- The forward-looking research agenda and the epistemic partnership vision raise focused and generative questions. They move beyond critique to help shape future work on uncertainty-aware collaboration.


**Weaknesses**

- The framework offers limited guidance on concrete implementation. Key concepts like "epistemic legitimacy" remain abstract. The paper does not propose metrics, representation strategies, or learning signals for tracking cognitive states.

- As a position paper, the work lacks case studies, simulations, or illustrative prototypes. A minimal proof-of-concept or a re-analysis of existing agent behaviors through the coupling lens would make the argument more persuasive.

- The distinction between "epistemic partnership" and existing mixed-initiative or proactive clarification approaches could be clearer.

**Support:**

3

---

> ### Author Rebuttal · Authors · 2026-03-29
>
> We sincerely appreciate the reviewer’s careful reading of our work, especially the attention to its **cross-disciplinary grounding**. We are glad that the paper is seen as identifying a **shared structural limitation in existing paradigms**, and that the proposed research agenda is viewed as useful for shaping future work on proactive agents.
>
> ### **Weaknesses**
>
> > *Operationalizing epistemic legitimacy through measurable signals and learning objectives.*
>
> We agree that epistemic legitimacy can appear abstract if read as a high-level normative concept. In practice, epistemic states are not directly observable but can be approximated through measurable signals of incomplete or unstable understanding (inconsistency across similar inputs, reliance on unverified assumptions, or repeated need for clarification). These signals both indicate epistemic gaps and provide criteria for system behavior. As outlined in **Section 5**: when such indicators suggest incomplete understanding, the system should reduce commitment; when signals are stable, it can act more decisively. This envisions the epistemic–behavioral coupling as policy decisions conditioned on epistemic signals.
>
> In **Appendix B (B.2–B.3)**, we further connect these ideas to learning and evaluation. For example, training objectives can discourage overconfident actions under incomplete information and reward clarification or deferral, while evaluation can assess whether actions were justified given the information available at the time, rather than only by final outcomes. These signals can be incorporated as features or reward components in learned policies (e.g., RL-based systems).
>
> > *Illustration through examples or prototype analyses*.
>
> We agree that concrete illustrations would make the framework more tangible. Our goal in this position paper is to first clarify the underlying gap, namely, that current systems do not explicitly reason about *missing or unmodeled aspects of a problem*, before committing to a specific implementation. At the same time, emerging work [1.] provides early evidence in this direction. Consider example [1.]:  *Query: "Because I have type 1 diabetes, what is likely to happen to me if I get the coronavirus?"*
>
> | System                | Response                                                                                                                                                                                                                                                                                                                                                |
> | - | - |
> | **GPT-4**             | “If you have type 1 diabetes and get COVID-19, you’ll most likely have mild symptoms and recover, especially if you’re young. The main issue is that illness can cause higher blood sugars and ketones, so you may need more insulin and closer monitoring. Serious problems are uncommon when diabetes is well managed.” |
> | **PROPER** | “Because you have Type 1 diabetes, COVID-19 might be more severe. Your immune system may struggle, raising the risk of breathing complications, and it can affect blood sugar control. Vaccination helps, avoid crowds, practice hygiene, and follow your doctor’s advice. Other conditions like hypertension, kidney, or heart disease increase risk.” |
>
> Both responses are helpful, but they differ in an important way. The first stays within the user’s framing and provides reassurance and guidance. The second surfaces *additional, unasked-but-relevant factors* (e.g., comorbidities, preventive actions), effectively expanding the problem space. We view this as an early stage instance where behavior is guided by reasoning about what might be missing from it.
>
> [1.] Gupta. et al. The PROPER Approach to Proactivity: Advancing Knowledge Gap Navigation.
>
>
>
> > *Distinction between "epistemic partnership" and existing mixed-initiative or proactive clarification.*
>
> We agree that this distinction should be made sharper. Our intention is not to reject mixed-initiative or clarification-based approaches; indeed, the paper explicitly recognizes them as important advances in interactive and collaborative AI. These approaches typically regulate how to act within an assumed task frame: when to interrupt, when to clarify, how to share control, or how to improve coordination once the relevant problem structure is already sufficiently specified. **Epistemic partnership**, by contrast, is aimed at cases where the framing itself may be incomplete. As **Section 6 and the Alternative Views** discussion argue, the key issue is not simply asking more questions or increasing interaction, but surfacing latent epistemic gaps, articulating missing relationships, preserving uncertainty long enough for discovery, and calibrating initiative by epistemic legitimacy. In that sense, epistemic partnership is not merely richer interaction; it is a governing constraint on how proactive behavior should unfold under epistemic incompleteness.

---

> > ### Author Rebuttal · Reviewer_DPN2 · 2026-04-02
> >
> > The authors' responses are agreeable. As the current manuscript and rebuttal align with my initial understanding of the work, the original score is maintained.

---

### Decision · Program_Chairs · 2026-04-30

**Decision:**

Accept (regular)

**Comment:**

The overall review is clearly positive. The main strengths are the paper’s strong conceptual framing, useful cross-disciplinary synthesis, and a research agenda that gives the community a richer vocabulary for reasoning about proactivity under epistemic incompleteness. The remaining concerns are mostly about empirical grounding, implementation detail, and whether some of the motivating claims about current systems are stated too broadly, but they are not fatal to the positional nature of the paper.